# ATP-Sensitive Potassium Channels in Migraine: Translational Findings and Therapeutic Potential

**DOI:** 10.3390/cells11152406

**Published:** 2022-08-04

**Authors:** Amalie Clement, Song Guo, Inger Jansen-Olesen, Sarah Louise Christensen

**Affiliations:** 1Glostrup Research Institute, Department of Neurology, Danish Headache Center, Copenhagen University Hospital—Rigshospitalet, Nordstjernevej 42, Glostrup, 2600 Copenhagen, Denmark; 2Department of Odontology, Panum Institute, Faculty of Health, University of Copenhagen, 2300 Copenhagen, Denmark

**Keywords:** K_ATP_ channels, provoked migraine, SUR, Kir6.x, levcromakalim, glibenclamide, human migraine model, in vivo models, migraine

## Abstract

Globally, migraine is a leading cause of disability with a huge impact on both the work and private life of affected persons. To overcome the societal migraine burden, better treatment options are needed. Increasing evidence suggests that ATP-sensitive potassium (K_ATP_) channels are involved in migraine pathophysiology. These channels are essential both in blood glucose regulation and cardiovascular homeostasis. Experimental infusion of the K_ATP_ channel opener levcromakalim to healthy volunteers and migraine patients induced headache and migraine attacks in 82-100% of participants. Thus, this is the most potent trigger of headache and migraine identified to date. Levcromakalim likely induces migraine via dilation of cranial arteries. However, other neuronal mechanisms are also proposed. Here, basic K_ATP_ channel distribution, physiology, and pharmacology are reviewed followed by thorough review of clinical and preclinical research on K_ATP_ channel involvement in migraine. K_ATP_ channel opening and blocking have been studied in a range of preclinical migraine models and, within recent years, strong evidence on the importance of their opening in migraine has been provided from human studies. Despite major advances, translational difficulties exist regarding the possible anti-migraine efficacy of K_ATP_ channel blockage. These are due to significant species differences in the potency and specificity of pharmacological tools targeting the various K_ATP_ channel subtypes.

## 1. Introduction

According to the World Health Organization (WHO), more than a billion people are living with migraine, and among the 15–49 year-old population, headache disorders is the most burdensome of all disorders [1,2]. Migraine attacks are characterized by pulsating head pain of moderate to severe intensity, photo- and/or phonophobia, nausea, vomiting, and aggravation by routine physical activity [3]. Migraine has a tremendous impact on quality of life for sufferers and may affect sleep [4], cognitive function [5], and private and professional life [6]. Despite huge individual suffering and socioeconomic impact, the pathophysiological mechanisms of migraine remain incompletely understood and highly debated [7]. The brain is generally thought of as non-nociceptive, but plexuses of nociceptive nerve fibers from the trigeminal ganglion innervate the blood vessels of the meninges (dura, pia, and arachnoid mater), linking pain perception and the brain vascular system in what is described as the trigeminovascular system [8]. Nowadays, many think of migraine as a neurovascular [7] sensory threshold disease [9]. The identification of calcitonin gene-related peptide (CGRP) involvement in migraine is a translational success story culminating in the marketing of monoclonal antibodies targeting CGRP or its receptor as well as small molecule receptor antagonists [10]. However, these CGRP-targeting migraine preventatives are only effective in approximately 60% of patients [11,12,13,14,15], stressing the importance of continued research and drug development.

A range of migraine-provoking substances have been identified in human experiments. Common for all, is the dilation of cephalic arteries [16,17,18,19,20,21] via downstream opening of vascular smooth muscle ATP-sensitive potassium (K_ATP_) channels [7]. The finding that the K_ATP_ channel opener levcromakalim is the most potent trigger of experimental migraine tested to date [22,23] has fueled interest in K_ATP_ channel involvement in migraine pain generation and, within recent years, a significant number of studies have addressed this topic.

The aim of this review is to collectively present the evidence on K_ATP_ channel involvement in migraine pain and review the underlying hypotheses of where and how the K_ATP_ channels are involved in migraine pathophysiology. Logically, the possibility of targeting antimigraine therapeutics against these channels will also be discussed.

## 2. Molecular Basis and Physiological Function of K_ATP_ Channels

### 2.1. Molecular Structure and Regulation of Channel Activity

The K_ATP_ channels were first identified in cardiac muscle cells by A. Noma in the early 1980s [24]. These channels were later shown in other tissues, such as pancreas, smooth muscle cells, and the nervous system [25,26,27]. They belong to the family of transmembrane potassium inward-rectifying (Kir) channels, which are predominantly found on the plasma membranes but are also present on the mitochondrial inner membrane [28]. Seven subfamilies within the Kir family have been identified with different molecular and physiological functions (Kir1.x through to Kir7.x), where ATP-sensitive K^+^ channels belong to the Kir6.x subfamily and are strongly associated with cellular metabolism and membrane electrophysiology [27]. Kir6.x have two subtypes, namely Kir6.1 and Kir6.2, which are expressed in various tissues [29].

Kir channels have two transmembrane spanning regions (TM1 and TM2) with an extracellular pore-forming region (H5) and both the amino and carboxyl terminal are cytosolic (Figure 1A) [29,30]. However, to obtain a functional channel, four Kir subunits are necessary, and the activity of the channel is regulated by four sulfonylurea receptors (SUR), thus creating a hetero-octameric structure (Figure 1B) [29,31]. These SUR receptors are ATP-binding cassettes (ABCs) or transport ATPases and have 17 transmembrane regions arranged into three domains (TMD0, TMD1 and TMD2) together with two intracellular nucleotide binding domains (NBD1 and NBD2); see Figure 1A. SUR subunits SUR2A and SUR2B only differ at the carboxyl terminal 42 amino acids (C42), while the SUR1 subunit is more unique [27,29,30].

The inward-rectifying function is a result of an intracellular blockage of the pore by Mg^2+^ or polyamines, which blocks the efflux of K^+^. During channel activation the blockage is removed and K^+^ efflux can occur [27]. High concentrations of ATP will inhibit the channel, while reduced ATP levels will activate and open the channel [32]. The activity of the channel is controlled by the SUR subunits due to their NBDs, where MgATP binds to NBD2 and MgADP binds to NBD1 [29,33]. Phosphatidylinositol 4,5-bisphosphate (PIP_2_) is suggested to play a role in channel activity regulation, as PIP_2_ activates the channel and reduces its sensitivity to ATP, thus counteracting the inhibitory effect at ATP [34,35,36]. Lastly, Kir6.x channels can be activated via phosphorylation by protein kinase A (PKA) or protein kinase G (PKG) [37,38,39,40]. These kinases are downstream targets of cAMP and cGMP, respectively, and have been suggested as molecular pathways in migraine pathophysiology (Figure 2) [41,42].

### 2.2. Tissue Distribution

The K_ATP_ channels are expressed throughout the body but the combination of the different subunits of Kir6.x and SURx vary in different tissues, such as the vascular system, neuronal system, and pancreas (see Table 1). The pancreatic β-cells express Kir6.2/SUR1, which control the glucose-stimulated insulin secretion (and represents the most studied channel), while the Kir6.2/SUR2A channels are the predominant form found in myocardia [25,29,33]. The vascular smooth muscle cells express Kir6.1/SUR2B and these have distinct structural features from the pancreatic Kir6.2/SUR1 isoforms, as the Kir6.1 cytoplasmic regions is placed too far from the membrane to interfere with the membrane-bound PIP_2_, which is known to activate or open the Kir6.2/SUR1 channels in pancreatic β-cells [33]. Furthermore, Kir6.1 channels do not show spontaneous channel activity, while pancreatic and myocardial Kir6.2 channels open spontaneously when ATP levels are low or absent [33,38,43]. In most tissue, channels are composed of two homogenous Kir subunits and four homogenous SUR subunits; however, examples of more heterogenous compositions have been reported [44]. The different compositions of K_ATP_ channel subunits in different tissues potentially allow for more specific therapeutic targets in the development of novel drug candidates for specific pathologies.

Kir6.1/SUR2B is found in the smooth muscle cells of the vascular system, and are the dominant form in brain arteries and dura mater [45], where they are involved in vasodilation and constriction. For this reason, Kir6.1/SUR2B have been suggested as a target for migraine pain intervention [46,47].

### 2.3. Physiological Functions of K_ATP_ Channels

In a physiological resting state, K_ATP_ channels are blocked but allow a small inward current of K^+^, while the active or open state of the channel results in the efflux of K^+^, resulting in hyperpolarization of the membrane. Below, the physiological consequence of this in different cell and tissue types is presented.

#### 2.3.1. Vascular System

The tone of the vascular system is controlled by a sophisticated relationship of molecular functions causing vasoconstriction or vasodilation. K_ATP_ channels have long been known as the target of vasodilatory drugs like diazoxide and pinacidil [63] and their role in vasodilation have likewise been studied for decades [33,38,63,64,65].

In vascular smooth muscle cells, hyperpolarization caused by K^+^ efflux upon K_ATP_ channel opening will cause the inhibition of voltage-operated Ca^2+^ channels (VOCC), reducing Ca^2+^ influx and consequently causing smooth muscle relaxation and vasodilation (Figure 2) [33,66,67,68]. Many vasodilating substances target receptors or second messengers upstream from K_ATP_ channels. Nitric oxide (NO) binds to guanylyl cyclase (GC), which in turn converts GTP to cGMP, and cGMP can subsequently phosphorylate and open the K_ATP_ channels [68,69,70]. Additionally, K_ATP_ channels, located in the endothelium, mediate vasodilation to some extent [71]. The potent vasodilators CGRP and pituitary adenylate cyclase-activating peptide (PACAP) bind to their respective G-protein coupled receptors on vascular smooth muscle to activate the adenylyl cyclase (AC) enzyme, causing cAMP to be converted from ATP [40,72,73]. In addition, inhibitors of phosphodiesterase (PDE) type 3 and 5 (cilostazol and sildenafil) are marketed for their vasodilating effect for different indications [74,75]. PDEs degrade cAMP and cGMP; thus, the inhibition of PDEs causes the accumulation of cAMP and/or cGMP and downstream phosphorylation of the K_ATP_ channel [7]. Interestingly, all these drugs or substances have headache as a primary side effect [16,17,74,75,76]. Thus, the above mentioned mechanisms have been speculated to be important in migraine pathophysiology.

Genetic manipulations of specific subunits of the functional K_ATP_ channel may result in vascular issues like knockout of the *Kcnj8*, the gene that encodes Kir6.1, which was shown to cause sudden early death associated with an atrioventricular blockage that could not be rescued by the K_ATP_ channel opener pinacidil but was worsened by the vasoconstrictive agent methylergometrine [77]. A later in vivo study showed that the selective deletion of Kir6.1 in vascular smooth muscle cells resulted in hypertension and a loss of response to pinacidil but did not cause sudden death [64]. Thus, sudden death was likely related to the global deletion of Kir6.1. It is important to keep in mind that global knockout of specific genes might cause unexpected phenotypic traits due to a lack of expression of the gene during embryonic development and consequences thereof, which might not be expected if the gene had been expressed in the embryonic stages and silenced later in life [78,79,80].

#### 2.3.2. Neuronal Function

K_ATP_ channels (Kir6.2/SUR1, Kir6.2/SUR2A, and Kir6.2/SUR2B) are also expressed in the brain, especially in neurons where their activation causes hyperpolarization and reduced excitability [27,29,30], which is often related to a reduction in neurotransmitter release [81,82]. Hyperpolarization may lead to the activation of hyperpolarization-activated cyclic nucleotide-gated (HCN) channels of which the consequence is dependent on the given situation [83]. The excitatory neurotransmitter glutamate will cause an influx of Ca^2+^, which can result in an elevation in mitochondrial Ca^2+^ levels, depolarization of the mitochondria, loss of oxidative phosphorylation, ATP depletion, swelling and rupture of mitochondrial membrane and, subsequently, release of pro-apoptotic species, ultimately leading to neuronal cell death [81,84]. This cascade can be blocked or regulated by activation of the mitochondrial K_ATP_ channels due to their ability to cause hyperpolarization and reduced glutamate release, thus playing a role in neuronal protection from excitatory toxicity [81]. Moreover, neuronal mitochondrial K_ATP_ channels regulate intracellular Ca^2+^ concentrations during hypoxia, thus protecting the neuron from hypoxia [84]. The K_ATP_ channel opener, iptakalim, was used to rescue stress-induced mitochondrial damage and alleviate a depressive-like phenotype in the rodent chronic mild stress model of depression [85]. Altogether, activating K_ATP_ channels in neurons most likely serve a neuroprotective function during stressful stimuli like ischemia and oxidative stress [48,57,86].

#### 2.3.3. Analgesia, Antinociception, and Opioid Signaling

K_ATP_ channels are, furthermore, involved in opioid signaling, which appears to be selective to these types of channels as no other K^+^ channels have been illustrated to bear similar properties [87]. Nevertheless, other K^+^ channels are implicated in pain perception but are beyond the scope of this review. For an extensive review on K^+^ channels in the pathophysiology of pain, please see [88].

The K_ATP_ channels have been shown to be downstream targets of the opioid receptor via regulation of nitric oxide synthase and NO generation and subsequent efflux of K^+^ and hyperpolarization [89]. This NO/cGMP/K_ATP_ channel pathway has further been suggested to be an important factor in the inflammatory nociceptive response [70,90,91]. Additionally, loss of K_ATP_ channels or their function in peripheral neurons has been implicated in pain perception in multiple studies [57,58,92,93]. For instance, expression of Kir6.1, SUR1, and SUR2 in rat spinal cord were downregulated after nerve injury [94] and knockout of SUR1 subtype resulted in the loss of the antinociceptive effect of morphine in mice [95,96]. Furthermore, the expression of Kir6.1/SUR2B were shown to be regulated by the inflammatory toll-like receptor 4 and NF-κB-dependent signaling, which was suggested to be a factor in the poor vasoconstriction during sepsis [97]. Furthermore, the K_ATP_ channel opener, cromakalim administered centrally, reduced astrocyte activation and lowered expression of IL-1β and TNF-α, thereby reducing neuroinflammation [98], and relieved injury-induced neuropathic pain both acutely (lasting hours) and chronically (lasting days) [94]. Khanna et al., 2011 showed the analgesic effect of the systemic administration of cromakalim in the formalin test to the same degree as morphine but only high doses of cromakalim (1 mg/kg and 5 mg/kg) induced an analgesic effect in the tail flick test [87]. Combining morphine and cromakalim only showed additive analgesic effects in the formalin test (inflammatory pain) and not in the tail flick test (heat sensitivity) [87]. Likewise, the K_ATP_ channel blocker, glibenclamide, increased the pain response to the formalin test but not in the tail flick test [87]. Interestingly, central administration of cromakalim did not induce an analgesic effect in the tail flick test [98]. One could speculate that the reason for this is the lack of inflammatory agents in the tail flick test, which would suggest that the antinociceptive effect of K_ATP_ channel openers is related to anti-inflammatory properties, resulting in analgesia in inflammatory pain rather than neuropathic pain.

Overall, the involvement of K_ATP_ channels in analgesic or antinociceptive mechanisms appear to be closely related to their ability to hyperpolarize the membrane and reduce hyperexcitability induced by inflammatory events. However, even though K_ATP_ channels in pain research have been studied for decades, no therapeutic agent has reached the market, and we speculate that the reason for this lack of drug development may be due to the high complexity of these mechanisms, the abundant distribution of K_ATP_ channels throughout the body and lack of subtype specific pharmacological tools.

Interestingly, when levcromakalim (K_ATP_ channel opener) was systemically administered, humans developed headache or migraine [22,23] and mice became hypersensitive [45,47,99]. When levcromakalim was centrally administered in mice, analgesic effects became evident [45,100]; see Section 4. (K_ATP_ Channels and Headache).

#### 2.3.4. Insulin Secretion and Glucose Metabolism

The pancreatic islet β-cells are responsible for the secretion of insulin, and Kir6.2/SUR1 K_ATP_ channels play an important role in this mechanism. When blood glucose levels rise, cell metabolism increases, and the higher ATP levels will trigger the closing of K_ATP_ channels, resulting in membrane depolarization and the activation of voltage-gated Ca^2+^ channels and Ca^2+^ influx, which ultimately leads to insulin secretion [101,102]. It is generally accepted that the loss of function of K_ATP_ channels in islet β-cells leads to poor or no response of K_ATP_ channels to changes in ATP/ADP ratio, leaving the channels closed and the membrane at a depolarized state. This leads to a rise in cytosolic Ca^2+^ and the secretion of insulin, causing the phenotype of hyperinsulinism [103,104]. In contrast, gain-of-function mutations can lead to reduced or absent secretion of insulin, hyperglycemia, and diabetes due to a high degree of K^+^ efflux and membrane hyperpolarization [102]. Congenital hyperinsulinism is linked to the loss of function of the Kir6.2/SUR1 K_ATP_ channel expression in the β-cells [101,104].

Type 2 diabetes mellitus is linked to chronic low-grade systemic inflammation and oxidative stress in β-cell, leading to low or no insulin secretion due to a loss of β-cells [105,106]. Several studies have reported that reactive oxygen species (ROS) and reactive nitrogen species (RNS) modulate K_ATP_ channel activity by inhibition of mitochondrial ATP production [107,108,109,110,111]. One study illustrated that oxidative stress-induced loss of β-cells caused high blood glucose levels in wild-type (WT) mice, while in SUR1^−/−^ mice, glucose levels only rose slightly over control levels and these mice had a significantly better survival rate compared to WT mice [107]. As the SUR subunit of the K_ATP_ channel holds the nucleotide binding sites, this illustrates that the mitochondrial ATP production is involved in the negative effects of oxidative stress on the insulin secretion pathway. One could speculate that oxidative stress in other tissues also disturbs the K_ATP_ channel function, for instance, in the migraine-relevant trigeminovascular system [112,113].

## 3. Pharmacological Tools Targeting K_ATP_ Channels

Pharmacological investigation using the different and more or less selective openers and inhibitors of the K_ATP_ channel is crucial for the understanding of channel function and interpretation of results. Both categories include several chemical classes. In summary, there is some selectivity of the K_ATP_ channel openers and blockers, but one should keep in mind that selectivity is commonly not an all or nothing phenomena but a function of dose.

### 3.1. K_ATP_ Channel Openers

Levcromakalim or cromakalim belongs to the benzopyran class and primarily opens smooth muscle and cardiomyocyte K_ATP_ channels via affinity to TMD2 on SUR2 [48]. PK_i_ values for levcromakalim is 6.37 ± 0.04 on SUR2A and 6.95 ± 0.03 on SUR2B [114]. In vasomotor experiments, levcromakalim has a reported pEC_50_ of 6.36 ± 0.09 on human pial arteries and 6.32 ± 0.3 on omental arteries [115]. In rats, similar studies found values of 6.32 ± 0.09 and 5.46 ± 0.17 for levcromakalim on basilar and middle cerebral arteries, respectively [116], and 7.14 ± 0.11 on middle meningeal arteries [117].

The cyanoguanidines (pinacidil, P-1075) also display selectivity to SUR2 and have potent hypotensive effects [118,119], but like levcromakalim, pinacidil did not reverse glibenclamide-induced SUR1-mediated hyperglycemia [120].

In contrast, diazoxide (benzothiadiazine) is more specific for SUR1, but also activates SUR2B [121]. Accordingly, it displayed both hypotensive and hyperglycemic effects [120,122,123].

### 3.2. K_ATP_ Channel Blockers

The sulfonylureas (glibenclamide, glicazide, and gliquidone) are compounds that target the SUR subunits to inhibit K_ATP_ channels via NBD2 (Figure 1A) [124]. Sulfonylureas are used clinically to promote insulin secretion in type 2 diabetes [125]. Glibenclamide has a 50-fold higher affinity towards SUR1 over SUR2A and SUR2B [126], and the inhibition of Kir6.2/SUR1 is poorly reversed, whereas the blocking of Kir6.2/SUR2A is rapidly reversed [127]. In addition, gliquidone and glicazide display lower IC_50_ values in pancreatic cells than in cardiomyocytes and vascular smooth muscle, suggesting a higher affinity for SUR1 [128]. Species differences are reported on the activity of glibenclamide on the human and mouse Kir6.1/SUR1 channels. pIC_50_ is 8.37 on the human and 5.7 on the mouse channel [126,129] suggest a much larger potency of glibenclamide on human compared to mouse pancreatic K_ATP_ channels.

The thiazolidinediones (rosiglitazone, pioglitazone, etc.) are another class of drugs used clinically for the treatment of type 2 diabetes via their effect on peroxisome proliferator-activated receptors (PPARs) [125]. However, patch clamp experiments on HEK cells expressing various subtypes of K_ATP_ channels have shown that these compounds also block vascular K_ATP_ channels to various degrees [130]. Rosiglitazone was found to inhibit all isoforms of K_ATP_ channels. The IC_50_ was 10 µM for the Kir6.1/SUR2B channel and ∼45 µM for KIR6.2/SURx channels. The inhibition was also present without the SUR subunit. Additionally, rosiglitazone had no effect on Kir1.1, Kir2.1, and Kir4.1 channels, suggesting that the channel inhibitory effect is selective for Kir6.x channels [131]. In a subsequent study, the same group compared a large number of PPAR agonists and found some to potently inhibit Kir6.1/SUR2B. The most potent agonists were AS-252424, englitazone, A6355, rosiglitazone, and cay10415 with IC_50_ values of 4 μM, 7 μM, 8 μM, 12 μM, and 15 μM, respectively. Lastly, the morpholinoguanidine PNU-37883A needs mentioning. Both in vitro and in vivo evidence suggest that this compound is selective for vascular K_ATP_ channels over pancreatic ones and that it acts on the Kir6.1 rather than SUR components [132].

## 4. K_ATP_ Channels and Headache

Clinical trials with K_ATP_ channel openers for the treatment of hypertension and asthma had headache as a primary side effect [63,133,134]. A range of migraine- and headache-triggering substances (NO, CGRP, PACAP, cilostazol, sildenafil, and more) activate K_ATP_ channels downstream from target binding (Figure 2). This agrees with the theory that the arteries of the trigeminovascular system, are involved in the generation of migraine pain [7].

### 4.1. Levcromakalim Is a Potent Trigger of Experimental Headache and Migraine

In an experimental setting, recognized as the human model of migraine [135], intravenous infusion of levcromakalim (1 mg over 20 min) was studied in healthy volunteers [136], including migraine without aura (MO) patients [22] and migraine with aura (MA) patients [23].

After levcromakalim infusion, 12 of 14 healthy volunteers reported a headache with a median time to onset of 30 min (range 10–60 min) compared to 1 of 6 participants after placebo [136]. In MO patients, migraine attacks without aura were induced in 16 out of 16 patients contra 1 out of 16 participants after placebo [22]. The median time to migraine onset was 3 h (range 1-9 h) after levcromakalim infusion. In MA patients, attacks were induced in 14 out of 17 participants, and 1 participant after placebo. Four attacks were MO whereas ten were MA. Median time of onset for MO was 2.8 h (range 1–4 h) and 44 min (range 20–120 min) for MA [23].

Apart from headache characteristics, hemodynamic parameters and circumference or blood flow velocity of selected arteries were reported in the above-mentioned clinical studies. In healthy participants, the middle meningeal artery (MMA) had a 7–22% larger circumference throughout the 5 h test period measured with 3.0 Tesla (magnetic resonance angiography (MRA)). The superficial temporal artery (STA) also dilated, but less robustly throughout the test period. The middle cerebral artery (MCA) dilated but it was not significantly different from placebo. Heart rate (HR) AUC_0–290_ was significantly increased but mean arterial blood pressure (MAP) AUC_0–290_ was not significantly lowered [136]. However, in a larger study, all arteries (STA, MMA, and MCA) dilated, and HR and MAP also changed significantly in response to levcromakalim [137]. In MO and MA patients both HR AUC_0–120_ and MAP AUC_0–120_ were also significantly altered. In the MO patient study [22], STA and radial artery diameters were measured by ultrasonography and blood flow velocity of the MCA was measured by transcranial doppler as a proxy for arterial circumference. Thus, this method is inferior to MRA. Only the STA was found to dilate in response to levcromakalim. Effects on arterial dilation was not published in the MA study [23].

### 4.2. K_ATP_ Channel Opening in Preclinical Migraine Models

Preclinical models in which the effect of K_ATP_ channel opening and inhibition have been studied in the context of migraine include models of vasoactivity, CGRP release, mast cells degranulation and behavior in rodents.

#### 4.2.1. Dilatory Effects on Cranial Arteries

The effect of K_ATP_ channel opening on cranial arteries have been studied both ex vivo using the wire myograph technique and in vivo using intravital microscopy through a closed cranial window in anesthetized rats. The latter allows simultaneous imaging of dural and pial arteries in an intact animal. It was found that levcromakalim infusion (0.1 mg/kg i.v.) increased dural artery diameter by 130 ± 24%, pial artery diameter by 18 ± 3%, and lowered MAP by 31% [138]. For pinacidil (0.38 mg/kg i.v.), the figures were 126 ± 8% and 17 ± 3%, respectively. The response was significantly lower in pial than in dural arteries for both K_ATP_ channel openers [138]. A lower dose of levcromakalim (0.025 mg/kg i.v.) sub-maximally dilated the MMA and decreased MAP by 29% [117]. In ex vivo artery preparations, the picture was similar. Levcromakalim (3 µM) induced relaxation of 74 ± 9% in rat dural arteries and 38 ± 8% in middle cerebral arteries, and pinacidil (3 µM) induced relaxation that amounted to 55 ± 11% and 26 ± 4%. Again, the dilatory responses were significantly different between the dural and cerebral arteries [138]. This is also reflected in the pEC_50_ values; see Section 3.1. (K_ATP_ Channel Openers), revealing an approximate 10-fold higher potency of levcromakalim on meningeal over cerebral arteries [117,139]. The in vivo findings could be explained by poor blood-brain barrier passage of levcromakalim and pinacidil, but this cannot explain the ex vivo difference as the blood-brain barrier is bypassed in the wire myograph technique. Therefore, it was suggested that K_ATP_ channels are heterogeneously distributed between cranial arteries [138].

#### 4.2.2. Stimulation of CGRP Release

The CGRP release assay is an ex vivo technique in which CGRP release from isolated tissue can be investigated. In migraine research, CGRP release from relevant structures as the trigeminal ganglia, trigeminal nucleus caudalis and dura mater are commonly investigated [140]. Levcromakalim (1 µM) and diazoxide (10 µM) have been tested for their ability to stimulate CGRP release from all three tissues in rats [141]. Levcromakalim (0.1–100 µM) was tested in mouse trigeminal ganglia and brain stem [99]. In neither species nor tissue preparation did levcromakalim induce release of CGRP, supporting the hypothesis that K_ATP_ channel opening is a downstream event upon CGRP receptor activation within single cells [142]. For review of the effect of K_ATP_ channel blockage on CGRP release, see Section 4.3.1. (Effect of K_ATP_ Channel Blockers in Preclinical Models). From the CGRP release model, it is evident that the headache-inducing effect of levcromakalim is not caused by direct stimulation of neuronal CGRP release.

#### 4.2.3. Mast Cell Degranulation

Dural mast cells may be involved in migraine attacks [143] and therefore mast cell degranulation assays and markers have been applied to study this possible aspect of headache mechanisms. Levcromakalim and diazoxide (10 µM) failed to degranulate rat dural mast cells in situ and, likewise, both drugs (0.01 µM–10 µM) failed to degranulate rat peritoneal mast cells in vitro [141]. Thus, degranulation of mast cells is not the primary mechanism in headache caused by levcromakalim.

#### 4.2.4. In Vivo Mouse Model

Many migraine triggering substances defined in the human migraine model have also been used in mice where they induce a state of hypersensitivity to cutaneous stimulation with von Frey filaments [99,144,145,146]. This model is considered to be the mouse parallel to the human model of provoked migraine. Repeated (every 48 h) injections of levcromakalim (1 mg/kg i.p.) induce both cephalic and hind paw hypersensitivity to von Frey stimulation, peaking 2 h after the 3rd injection [45,47,99], whereas levcromakalim administered locally in the hind paw did not induce hypersensitivity, and intracerebroventricular administration provided analgesia on the hotplate [45]. The observed hypersensitivity is at odds with the study by Khanna et al., 2011 showing an antinociceptive effect of cromakalim and diazoxide delivered i.p. [87].

### 4.3. K_ATP_ Channel Blockage as Therapeutic Target in Migraine

The opening of K_ATP_ channels by systemic levcromakalim induces headache and migraine attacks with and without aura. Accordingly, blocking K_ATP_ channels may abort migraine attacks. A convincing amount of preclinical evidence suggests that K_ATP_ channel blockage is a promising drug target for migraine. However, translation to patients is pending better pharmacological tools. Table 2 summarizes the studies reviewed in the following sections and provides an overview of the doses of applied test substances expressed as µmol/kg and the ratio between blocker (glibenclamide) and headache trigger substance.

#### 4.3.1. Effect of K_ATP_ Channel Blockers in Preclinical Models

The preclinical evidence suggesting the relevance of K_ATP_ channel blockage in migraine is based on evidence from studies of (a) cranial arteries, (b) CGRP release, (c) behavioral models, and (d) a genetically modified model:

(a) High dose glibenclamide (30 mg/kg i.v.) effectively blocked vasodilation in rats induced by levcromakalim and pinacidil in both dural and pial arteries in vivo [31,138]. Glibenclamide also inhibited dilation caused by migraine-triggering peptides CGRP [65] and PACAP [147] that support K_ATP_ channel activation by phosphorylation via cAMP and PKA [148]. PNU-37883A effectively inhibited dilatory responses to stimulation with K_ATP_ channel openers in various arteries of different species including the MMA in vitro and in vivo [117,132]. Interestingly, glibenclamide [65] and PNU-37883A [149] failed to inhibit arterial dilation caused by NO-donors in some reports whereas others did find a relationship between NO (cGMP) and K_ATP_ channel-mediated arterial dilation [45,150].

(b) Glibenclamide (3 µM) inhibited ex vivo capsaicin-induced CGRP release from trigeminal ganglia and dura mater from spontaneous trigeminal allodynic (STA) [151,152] rats via an unknown mechanism [47].

(c) Also, in the STA rat model of migraine, glibenclamide (1–10 mg/kg i.p.) and gliquidone (10-100 mg/kg i.p.) reversed spontaneous trigeminal allodynia [47]. In the mouse models of provoked migraine, glibenclamide (1 mg/kg i.p.) was highly effective against levcromakalim, cilostazol, and glyceryl-trinitrate (GTN)-induced tactile hypersensitivity [47,99], whereas it only partially blocked the effect of PACAP-38 [146].

(d) Mice lacking the Kir6.1 subunit in smooth muscle cells were less sensitive to CGRP [64], levcromakalim and GTN [45] induced vasodilation and hypersensitivity.

#### 4.3.2. Clinical Effect of K_ATP_ Channel Inhibition in Human Migraine Models

Glibenclamide (10 mg p.o.) has been tested against levcromakalim [137,153], CGRP [154], and PACAP-38 [155] induced migraine or headache in healthy volunteers. Glibenclamide was given 2 h prior to levcromakalim and CGRP infusions, but after the PACAP infusion. Headache data and hemodynamic measures were obtained. In all studies, subjects continuously received glucose to counteract the pronounced drop in blood glucose caused by glibenclamide. Overall, glibenclamide was found ineffective against both hemodynamic changes and headache induction after infusion of all three migraine triggering compounds.

The three above-mentioned studies were all cross-over studies but with variations in experimental design. In the levcromakalim study, NCT03886922 [137,153,156], three study arms were included: placebo-placebo, glibenclamide-placebo, and glibenclamide-levcromakalim. The study did not have a placebo-levcromakalim group, making the conclusions a bit distorted. In total, 12/15 participants (80%) reported headache after glibenclamide-levcromakalim, 5/15 (33%) after glibenclamide-placebo, and 1/15 (7%) following placebo-placebo. Thus, glibenclamide itself did not induce headache at a rate significantly different from placebo. To test if glibenclamide protected against levcromakalim-induced headache, comparison was made to a previous study showing headache induction in 12/14 of participants (86%) after levcromakalim versus 1/6 (17%) after placebo [136]. Hence, glibenclamide pretreatment did not inhibit headache development but, noteworthily, the median time to headache onset was 30 min (range 10–60) after levcromakalim infusion without pretreatment and 180 min (range 20–600) with glibenclamide pretreatment (*p* = 0.007) [153]. Glibenclamide did not influence HR, MAP, nor the circumference of neither STA,, MMA, nor MCA, which, in this study, all significantly changed in response to levcromakalim [137].

The study on CGRP and glibenclamide included two experimental groups in a randomized cross-over design: placebo-CGRP and glibenclamide-CGRP. The incidence of headache on the placebo-CGRP day was 19/20 (95%) vs. 14/20 (70%) on the glibenclamide-CGRP day (*p* = 0.06). Biologically, this was a 25% reduction in headache inductions, but power was set to detect 50% reduction in the study; thus, we cannot with certainly say if this finding happened by chance. Glibenclamide clearly did not influence CGRP-mediated changes on the hemodynamic parameters arterial diameter, HR, MAP, and facial skin blood flow [154]. Similar findings were obtained with glibenclamide as posttreatment when headache was induced by PACAP-38 [155]. Here, 19/20 participants (95%) reported headache compared to 18/20 (90%) on the placebo-PACAP day (*p* = 0.698).

## 5. Discussion

Direct comparison between human and animal experiments is not straight forward [157]. Apart from the rule of thumb conversion factor for doses based on body surface area to account for a generally faster metabolism in smaller animals [158], several other factors may be of importance. For the studies reviewed here, the difficulties concern: (1) different routes of administration and lack of pharmacokinetic data to safely interpret their impact, (2) different measuring endpoints, (3) a lack of evidence on the exact K_ATP_ channel distribution and expression levels in different tissues and species, and (4) the potency of test compounds on different channel subtypes and receptors across species. In Table 2, these are mentioned with the possible effect it may have on the conclusions.

### 5.1. K_ATP_ Channel Opening Has Similar Effect in Preclinical and Clinical Studies

Across species, K_ATP_ channel openers dose-dependently dilate arteries and decrease blood pressure [48,137,138]. The headache- or migraine-inducing effect of levcromakalim infusion in humans was modeled in the mouse model of migraine, but with major differences that need mentioning. Humans received a single dose of 0.014 mg/kg i.v. to induce headache or migraine [22,23,136], whereas mice received one, two, or sometimes three i.p. injections of 1 mg/kg before hypersensitivity to tactile stimuli was evident [45,47,99]. Thus, translation is complicated by different routes of administration and different measuring endpoints, and no common readout to which the other measures can be related to. Some degree of first pass metabolism following i.p. administration of levcromakalim is likely due to portal absorption [159], thus somewhat lowering the actual mouse dose. The net conclusion is that, in addition to hemodynamic effects, migraine-relevant nociceptive pathways are also replicated in the mouse model but at a higher dosing regimen.

### 5.2. Discrepant Results on K_ATP_ Channel Inhibition in Preclinical and Clinical Studies

Evidently, there has been a poor translation between preclinical and clinical studies looking at K_ATP_ channel blockage with glibenclamide in a migraine context, both on the hemodynamic parameters and headache readouts (hypersensitivity to tactile stimulation in rodents). These discrepancies may be explained by differences in dosing regimens, pharmacodynamic action of glibenclamide, subunit distributions, and trigger potency on various receptors.

#### 5.2.1. Discrepant Effect of Glibenclamide on Cranial Arteries

In animal models using intravital microscopy of dural arteries, high doses of glibenclamide (7–30 mg/kg i.v.) were needed to prevent arterial dilation and a decrease in blood pressure followed by levcromakalim (0.1 mg/kg i.v.), pinacidil (0.38 mg/kg i.v.) [137], and CGRP (0.3 µg/kg i.v.) [65]. Looking at the glibenclamide/levcromakalim dose ratio (Table 2), it was 174 for full blockage and 116 for partial blockage of dural artery dilation in rats [138]. In the human equivalent study, the ratio was 5.8 (glibenclamide given to non-fasting participants has a high oral bioavailability, therefore this ratio is not adjusted). For CGRP, the ratio was 178,968 for the partially (but non-significant) effective dose of glibenclamide, and 767,004 for the fully effective dose in rats [65]. In the human experiment, the ratio was 24,972–53,690 depending on whether the total or 1 min CGRP dose was used. In animals, glibenclamide has only been tested against PACAP-induced arterial dilation in vitro. Here, the dose ratio calculation would not be meaningful to compare to the clinical data. In summary, the relative dose of glibenclamide contra trigger (levcromakalim, CGRP) was much higher in the rat studies. Given the SUR1 preference of glibenclamide, the relative low dose of glibenclamide given in humans may explain the lack of an effect of glibenclamide on human cranial arteries expressing SUR2B, while the higher dose applied to rats was sufficient to also inhibit SUR2B.

#### 5.2.2. Discrepant Effect of Glibenclamide on Headache Measures

In terms of headache measures, the effectiveness of glibenclamide in mouse and rat models of migraine has not been seen in the human model of provoked migraine. In both rats and mice, glibenclamide (1 mg/kg i.p.) was sufficient to inhibit cutaneous tactile hypersensitivity, but in the human studies, 10 mg p.o. (0.14 mg/kg) was not convincingly effective. The rodent dose was 7 times higher than the human dose (assuming equal bioavailability), which is the typical conversion factor between rats and humans based on body surface area [158]. Looking at the glibenclamide/trigger ratio in the mouse model and the human model, we found that in mice, the ratio was 0.58 for levcromakalim and 45,891 for PACAP, the latter only partially preventing hypersensitivity. In the human studies, the ratios were 5.79 for levcromakalim and 1.42 for PACAP. Despite different routes of administration, we get a clear indication that the glibenclamide/PACAP ratio was smaller in the human study compared to the mouse study, which may explain the lack of translation between results. However, for levcromakalim, the ratio was 10-fold higher in the human experiment, which leaves no simple explanation for the lack of efficacy on the primary readout. Recall, however, that in this study [153], a comparison was made to a previous study [136]. A highly significant effect was found on a secondary output, median time to headache onset, which was 30 min after levcromakalim and 180 min after glibenclamide-levcromakalim, suggesting that glibenclamide was effective when plasma levels were high [153]. Glibenclamide has not been directly tested against CGRP in the mouse model of provoked migraine.

#### 5.2.3. Target Engagement

The 10 mg dose of glibenclamide applied in the human studies was clearly insufficient in terms of blocking vascular K_ATP_ channels (Kir6.1/SUR2B), but the effect on blood glucose was shown to indicate the efficient blocking of pancreatic (Kir6.2/SUR1) channels in line with SUR1 selectivity of glibenclamide described in Section 3.2. (K_ATP_ channel blockers). In rats, the applied (high) dose was able to block the vascular K_ATP_ channels. In humans, larger doses cannot be applied due to the adverse effect on blood glucose [153]. The effect of blood glucose was not reported in the rat studies looking at hemodynamics after glibenclamide 7–30 mg/kg, i.v. [65,138]. In another rat study, both glibenclamide 1 mg/kg and 10 mg/kg i.p. decreased blood glucose from 7 mmol/L (vehicle treatment) to 3 mmol/L 2 h post-treatment. In mice, acute injection of glibenclamide (10 and 30 µg/mouse) caused a rapid, dose-dependent drop in blood glucose levels from approximately 170 to 120 mg/dL, peaking at 60 min. The two highest concentrations of glibenclamide caused a similar marked reduction of fed blood glucose after an extended period, consistent with a saturated effect of the drug in vivo [160]. Glibenclamide 5 mg/kg/day (delivered by a subcutaneous minipump) did not affect blood glucose in vivo [161]. The pronounced effect on human (but not mouse) glucose level is likely due to the reported > 100-fold higher potency of glibenclamide on human SUR1 over mouse SUR1 [126,162].

The higher selectivity of PNU-37883A on the vascular channels compared to glibenclamide is evident when the inhibition of levcromakalim is related across two studies from the same laboratory using the same in vivo model to study dilation of the MMA [117,138]. Here, 1.3 µmol/kg of PNU-37883A partly inhibited 0.087 µmol/kg of levcromakalim-induced dilation (ratio 15) and 40.5 µmol/kg of glibenclamide partly inhibited 0.35 µmol/kg of levcromakalim (ratio 116).

In both rats and mice, the lower dose of glibenclamide (1 mg/kg i.p. = 2 µmol/kg, ratio 0.58) was effective in different behavioral models of migraine [47,99]. The human studies on headache prevention by glibenclamide following provocation with levcromakalim, CGRP, and PACAP were negative on the primary outcome. Nevertheless, partial efficacy may have been present in the former two experiments; see Section 4.3.2. (Clinical effect of K_ATP_ channel inhibition in human migraine models). Partial inhibition of CGRP and PACAP induced alterations may be expected as the downstream effect from both neuropeptides likely also involve the opening of other ion channels [163].

### 5.3. Possible Mechanism of Headache Induction and Prevention

Different theories about where and how the opening of K_ATP_ channels causes headache exist. These are further fueled by the non-clarified effect of channel inhibition on headache readouts. Collectively, the interpretation regarding subunit contribution to headache is difficult as rodent and, to some extent, human data suggest that glibenclamide may inhibit headache (hypersensitivity in rodents) to some degree, without the relevant effect on the vascular Kir6.1/SUR2B channel, which is the proposed mediator of levcromakalim-induced migraine [22]. An effect on neuronal Kir6.2/SUR2 channels is also a possibility [164], albeit smooth muscle Kir6.1 subunits were identified as important [45].

#### 5.3.1. Dilation of Meningeal Arteries

Dilation of intracranial arteries within the trigeminovascular system is the leading hypothesis of levcromakalim-induced headache and migraine [7,45,165]. Two proposed mechanisms are currently at play: (a) mechanical activation of trigeminal nociceptors by arterial dilation or (b) chemical activation of trigeminal nociceptors by high [K^+^] in the microenvironment between arteries and nerve endings [23]. These hypotheses need testing using a selective blocker of Kir6.1/SUR2B suitable for use in humans.

#### 5.3.2. Effect on CGRP Signaling

A few preclinical studies suggest that K_ATP_ channels may affect CGRP signaling in different manners. In ex vivo organ preparations, glibenclamide inhibited the release of CGRP from trigeminal ganglia and dura mater [47]. In contrast, K_ATP_ channel openers did not directly stimulate the release of CGRP [99,141]. However, in vivo, the hypersensitivity induced by levcromakalim was abolished both by treatment with a CGRP-neutralizing antibody, and in genetically modified mice, by not expressing the CGRP receptor component RAMP1 [99], suggesting that CGRP is released by inter-tissue communication (not found ex vivo) following levcromakalim treatment and that this drives hypersensitivity. As an alternative to the vascular theory, this specific release of CGRP may in fact be what is inhibited by glibenclamide in vivo via its affinity to SUR1. This may also explain the speculative effect of glibenclamide on headache, in spite of the clear lack of a vascular effect on hemodynamics in human experiments.

#### 5.3.3. Hyperpolarization-Activated Cyclic Nucleotide-Gated (HCN) Channels

Another alternative to the vascular theory of migraine induction by K_ATP_ channel opening is the involvement of HCN channels in the trigeminal nervous system [164]. Sustained hyperpolarization of neurons may engage HCN channels, and blockage of these have been suggested as therapeutic targets in diabetic neuropathy [166] and neuropathic [167] and inflammatory pain [168]. The proposed mechanism is that K_ATP_ channel openers lead to the long-lasting hyperpolarization of trigeminal nerves, in turn activating HCN channels that leads to augmented neuronal excitability and firing of the neurons [169,170]. In contrast, this hypothesis consists of the fact that, in CNS, the opening of K_ATP_ channels is involved in analgesia. Moreover, HCN channels are expressed in trigeminal and dorsal root ganglia, and systemic exposure to levcromakalim induces cephalic but not peripheral pain [22,23,136] and local administration of levcromakalim was unable to induce pain [171]. The HCN theory is currently being evaluated in a clinical trial (NCT04853797), testing HCN channel blocker ivabradine against levcromakalim-induced headache [156].

### 5.4. Clinical Therapeutic Perspectives

Ion channels are regarded as an important class of drug targets for modulating pain and is localized in primary sensory neurons and other key structures in pain processing [172]. K_ATP_ channels are probably the most diverse ion channel type, and each subtype has a specific physiological role. Developing drugs targeting all K_ATP_ channels may therefore be impossible since they are widespread and undesirable severe side effects would be expected. Thus, subtype selectivity is key and may be a very attractive target for the development of novel therapeutics for the acute and preventive treatment of migraine.

Accordingly, Kir6.1/SUR2B subunits are dominantly expressed in the vascular smooth muscle. In contrast, Kir6.2/SUR1 are expressed in the CNS and pancreas, and Kir6.2/SUR2A are expressed in cardiac and skeletal muscle [50,117]. However, the lack of detailed structural and functional insight of these channels poses a challenge for the development of selective drug candidates. The Kir6.1 selective K_ATP_ channel blocker, PNU-37883A, was developed as an orally effective non-kaliuretic diuretic in rats [149,173]. Because of its cardiac depressant activity, possibly related to its blockade of coronary artery Kir6.1 channels in animal experiments, the drug never advanced to human studies [174]. Thus, K_ATP_ channel blockers selective for Kir6.1 alone should be carefully considered. In addition, SUR2 null mice exhibited elevated resting blood pressure and sudden death from ST segment elevation and coronary artery vasospasm [175]. In SUR2, in null mice with a transgenic restoration of SUR2B, the above-mentioned side effects persisted [176]. Thus, these side effects seem to be caused by SUR2A knockout and are likely not related to knockout of the SUR2B subunit. A K_ATP_ channel blocker for the treatment of migraine should therefore preferably have an exclusive selectivity for Kir6.1/SUR2B K_ATP_ channels.

To date, most ion channel drug development has focused on identifying and developing small molecule and peptide modulators, mainly through serendipitous discovery [177]. Despite vastly improved screening tools for small molecule or compound libraries, only two novel ion channel drugs have been approved by the FDA since the 1990s [178]. The well-known disadvantage of small molecules is that they can bind to off-molecular targets, leading to more side effects and toxicity. Alternative modalities for targeting ion channels have recently included monoclonal antibodies (mAbs), which offer many additional advantages to selectivity and bioavailability. Yet, despite considerable interest, there are currently no marketed mAbs therapies that target an ion channel. This lack of success is mainly attributable to two important technical challenges. First, the ion channels have short extracellular loops displaying small epitope target areas over the plasma membrane, causing them to be challenging binding targets for large protein antibodies. Additionally, these extracellular loops tend to be highly conserved at the primary amino acid sequence level, and thus lack sufficient immunogenicity to generate robust antibody responses in mammalian hosts [178].

A major challenge and concern in developing K_ATP_ channel blockers is cardiac side effects. K_ATP_ channels are abundant in the myocardium and K_ATP_ channel openers have proven useful in ischemic heart disease through direct actions on the myocardium [179] and may prevent arrhythmias [180]. To overcome this problem, it is important to test with several heart assays, such as the ex vivo Langendorff heart model (perfused isolated heart model), to evaluate the direct effects of compounds on cardiac function and to ensure cardiovascular safety of new drug candidates [181].

In conclusion, K_ATP_ channels are recognized as promising therapeutic targets for migraine treatment but remain a major challenge for drug discovery. To move forward, we need further studies on the specific subtypes of the K_ATP_ channel to enable a deeper understanding of their structures, functions and distribution for more selective and successful drug development. Furthermore, knowledge on the consequences of activation or blockage of the K_ATP_ channels on a molecular- or pathway-specific level in the pathophysiology of migraine, is necessary to fully comprehend and predict the potential of this novel therapeutic target.

## Figures and Tables

**Figure 1 cells-11-02406-f001:**
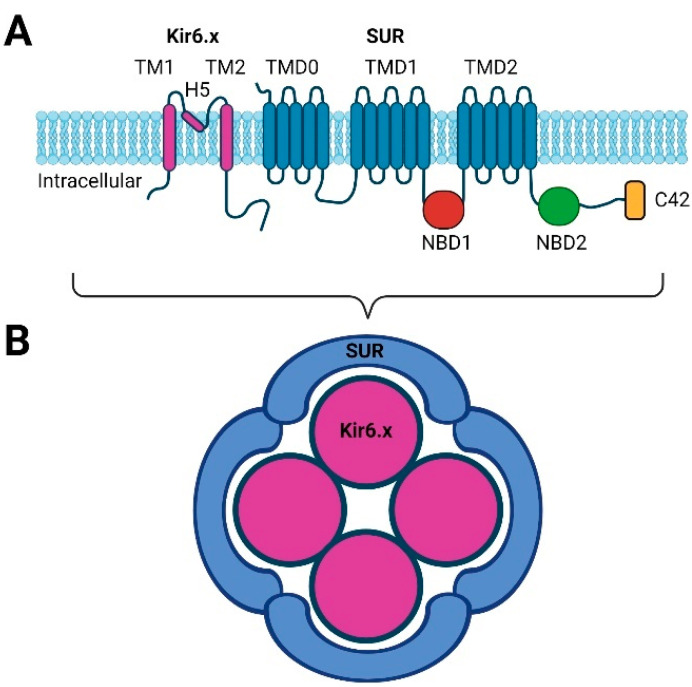
Simple structure of the K_ATP_ channel. (**A**) The Kir6.x subunit is composed of a two transmembrane region (TM1 and TM2) connected by a pore-forming region (H5). The SURx subunit is composed of three domains of either five transmembrane regions (TMD0) or six transmembrane regions (TMD1 and TMD2). The nucleotide binding domains are found intracellularly (NBD1 and NBD2). SUR2A and SUR2B only differ in their C-terminal end (C42). (**B**) The functional K_ATP_ channel is formed by four Kir6.x subunits and four SURx subunits (created using BioRender.com).

**Figure 2 cells-11-02406-f002:**
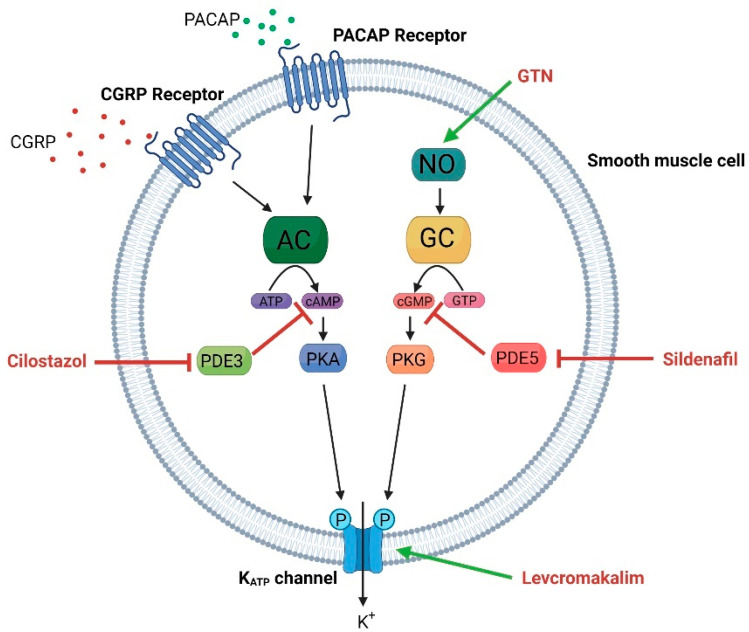
Molecular pathways and pharmacological agents leading to the opening of the K_ATP_ channel in vascular smooth muscle. The neuropeptides PACAP and CGRP activate K_ATP_ channels via the adenylyl cyclase pathway, while the NO donor GTN (glyceryl trinitrate) activates the channel via the guanylyl cyclase pathway. Cilostazol and Sildenafil are blockers of the phosphodiesterase 3 and 5 (PDE3 and PDE5), respectively, causing accumulation of cAMP and cGMP, which promote the opening of K_ATP_ channels. Levcromakalim causes vasodilation by direct action on the K_ATP_ channels (created using BioRender.com).

**Table 1 cells-11-02406-t001:** Subunits composition and tissue expression of K_ATP_ channels. For a more detailed overview of subunit composition, tissue distribution and physiological function, please see [48].

Channel Subunit Composition	Tissue	References
Kir6.1/SUR1	Retina	[49]
	Nervous system	[46,48]
Kir6.1/SUR2B	Vascular smooth muscle	[43,45,50,51,52]
	Non-vascular smooth muscle	[48,53]
	Conduction system of the heart	[48,54]
Kir6.2/SUR1	Pancreatic β-cells	[52,55]
	Arterial cardiac myocytes	[52,56]
	Nervous system	[48,52,57,58]
	Skeletal muscle	[48,59]
Kir6.2/SUR2A	Ventricular myocytes	[54,60]
	Skeletal muscle	[48,59]
Kir6.2/SUR2B	Non-vascular smooth muscle	[53]
	Nervous system	[48,57,61]
	Conduction system of the heart	[54,62]
	Skeletal muscle	[59]

**Table 2 cells-11-02406-t002:** Details of human and rodent studies on KATP channel blockage in different migraine models. Rows in same color are compared. The ratio of blocker/migraine trigger are used for rough assessment of effectiveness across models. Effective Y/N/P: Y = yes, N = no, P = partially. Percentwise changes of arterial circumference and diameter are the same. Thus, 20% change in diameter = 20% change in circumference. Dose mol/kg = (dose g/kg)/(MW g/mol), dose umol/kg = (dose mol/kg) × 10^6^. Glibenclamide 494 g/mol, levcromakalim 286 g/mol, PACAP 4534 g/mol, CGRP 3798 g/mol, PNU 382 g/mol. * Glibenclamide given after PACAP, # CGRP is accumulated dose in man/bolus in rat. Ratio will increase if the 1 min dose of CGRP is applied. $ Possible first pass metabolism of levcromakalim i.p will increase the mouse ratio, due to a smaller denominator. & PACAP s.c. may result in lower plasma concentrations than i.v. which will increase the mouse ratio.

Species	Endpoint	Headache Trigger mg/kg	Headache Trigger, umol/kg	Blocker mg/kg	Blocker, umol/kg	Ratio (Blocker/Trigger)	Effective Y/N/P
Rat	MMA diameter	Levcromakalim 0.025 mg/kg iv over 10 min	0.087	PNU-37883A 0.5 mg/kg i.v. over 10 min	1.3	15	P
Rat	MMA diameter	Levcromakalim 0.1 mg/kg iv over 20 min	0.35	Glibenclamide 20 mg/kg iv over 20 min	40.5	116	P
Rat	MMA diameter	Levcromakalim 0.1 mg/kg iv over 20 min	0.35	Glibenclamide 30 mg/kg iv over 20 min	60.7	174	Y
Human	MMA, STA, MCA circumference	Levcromakalim 0.014 mg/kg iv over 20 min	0.049	Glibenclamide 0.14 mg/kg p.o.	0.3	5.8	N
Rat	MMA diameter	CGRP 0.3 ug/kg iv bolus	0.000079	Glibenclamide 7 mg/kg iv over 20 min	14.2	178,968	P
Rat	MMA diameter	CGRP 0.3 ug/kg iv bolus	0.000079	Glibenclamide 30 mg/kg iv over 20 min	60.7	767,004	Y
Human	STA and RA diameter	CGRP 0.43 ug/kg iv over 20 min	0.000011	Glibenclamide 0.14 mg/kg p.o.	0.3	24,972 ^#^	N
Human	STA and RA diameter	CGRP 0.02 ug/kg/min i.v.	0.0000053	Glibenclamide 0.14 mg/kg p.o.	0.3	53,690	N
Human	MMA circumference	PACAP 200 picomol/kg over 20 min	0.2	Glibenclamide 0.14 mg/kg p.o. *	0.3	1.4	N
Human	Headache	Levcromakalim 0.014 mg/kg iv over 20 min	0.049	Glibenclamide 0.14 mg/kg p.o.	0.3	5.8	N/P
Mouse	Tactile hypersensitivity	Levcromakalim 1 mg/kg i.p ^$^	3.5	Glibenclamide 1 mg/kg i.p.	2	0.6	Y
Human	Headache	PACAP 200 picomol/kg over 20 min	0.2	Glibenclamide 0.14 mg/kg p.o.	0.3	1.4	N
Mouse	Tactile hypersensitivity	PACAP 0.2 ug/kg s.c. ^&^	0.000044	Glibenclamide 1 mg/kg i.p.	2	45,891	P
Human	Headache	CGRP 0.43 ug/kg iv over 20 min	0.000011	Glibenclamide 0.14 mg/kg p.o.	0.3	24,972	N/P

## Data Availability

Not applicable.

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
