# Peer review of "ATP-Sensitive Potassium Channels in Migraine: Translational Findings and Therapeutic Potential"

_cells, 2022, doi:10.3390/cells11152406_

Round 1

Reviewer 1 Report

This review is a well-edited and detailed manuscript on the role of ATP-sensitive potassium channels in migraine. The authors collected and assessed numerous references and produced a comprehensive summary from pharmacological, preclinical and clinical perspectives. Relevant information was incorporated into the manuscript and a logical structure was used. I support the publication of this review after some minor corrections.

My suggestions:

·       Abstract, Page 1: Please, correct these mistakes (erratum): lines 18 and 20: levcromaklaim and levcromakalaim; line 23: thourough; line 28: differnces

·       1. Introduction, Page 2: ATP sensitive potassium … - in the whole text, please standardize: ATP-sensitive (with hyphen)

·       2. Molecular basis and physiological function of KATP channels … Page 2: In the sub-title there is a mistake: „rugulation” – please, correct it.

·       2.3.1. Vascular system, Page 5: PACAP abbreviation was used without the whole name. Please, correct it.

·       2.3.2. Neuronal function, Page 6: line 185: Ca2+ - please, standardize in the whole text the 2+ in superscript (page 8, line 246, 247). 

·       2.3.4. Analgesia, antinociception and opioid signaling, Page 7: line 214 – I think there is an erratum: NK-kappaB – please correct it (NF). In line 219 – there is a missing dot: Khanna et al.

·       2.3.5. Insulin secretion and glucose metabolism, Page 8: line 247-253 – I think there is a too long sentence („It is generally accepted…” – please, split it). Also, in this section the „loss of function „ phrase was written with and without hyphen (line 248 and 254). In the line 260 you used the WT abbreviation without the explanation (wild type).

·       3.1 KATP channel openers, Page 8: line 285: hyperglecamia – please, correct it.

·       4.2.2. Stimulation of CGRP release, Page 11: line 390 – singe cells (maybe single?)

·       4.2.4. In vivo mouse model, Page 11: line 410 – ICV abbreviation is unnecessary

·       4.3.1. Effect of KATP channel blockers in preclinical models, Page 12: line 442 – the MMA abbreviation was used before, so please change the “middle meningeal artery” to MMA; page 13, line 453 and 456: the GTN abbreviation may not be used before – please check it.

·       4.3.2. Clinical effect of … … in human migraine models, Page 13: line 489, the heart rate was abbreviated before (HR on page 10, line 345), so you can use it.

·       Discussion, page 14, line 499: there is an erratum (typo): KTAP -please, correct it. Page 18: the FDA abbreviation is unnecessary. 

·       I think highlighting secondary captions would be useful to separate/break up the text (e.g. 4.2., because this title has several sub-titles: 4.2.1 and 4.2.2, etc). 

      I recommend to make a figure for the section of “4. KATP channels and headache” or the discussion, which may present the connections between migraine and ATP-sensitive potassium channels (similarities, discrepancies, targets, possible mechanisms, therapeutic perspectives – preclinical and clinical data). It would help in better understanding.

Reviewer 2 Report

This is an interesting and exhaustive review on KATP channel involvement in pathophysiology and on the possible utility of KATP channel as antimigraine therapeutic agents.

The paper is well written and covers a topic which is extremely interesting for the development of future migraine treatments.

1)    Migraine may occur without headache pain. This is the case of the so-called “Episodic syndromes that may be associated with migraine” (ICHD-III: 1.6) and of “Typical aura without headache” (ICHD-III: 1.2.1.2). Hence, the authors should acknowledge that KATP channel modulation has migraine consequences in most – but not in all – migraine patients

2)    The proportion of patients responding to antiCGRP mAbs is higher than 50%. Most prospective real-life studies document a >50% response in 60%-70% of migraine patients

3)    There are several typos in the abstract and throughout the paper

4)    Reference style should be checked (e.g., use of capital letters for authors/study title in ref #121,122, 142 and others
